# Potential of Higher Resolution Synchrotron Radiation Tomography Using Crystal Analyzer-Based Imaging Techniques for Differential Diagnosis of Human Lung Cancers

**DOI:** 10.3390/cancers18010082

**Published:** 2025-12-26

**Authors:** Eunjue Yi, Naoki Sunaguchi, Jeong Hyeon Lee, Miyoung Woo, Youngjin Kang, Seung-Jun Seo, Daisuke Shimao, Sungho Lee

**Affiliations:** 1Department of Thoracic and Cardiovascular Surgery, Korea University Anam Hospital, Seoul 02841, Republic of Korea; 2000180104@korea.ac.kr (E.Y.);; 2Photon Factory, Institute of Materials Structure Science, High Energy Accelerator Research Organization (KEK), Tsukuba 305-0801, Japan; naoki.sunaguchi@kek.jp; 3Department of Pathology, Korea University Anam Hospital, Seoul 02841, Republic of Korea; pathjhlee@gmail.com (J.H.L.);; 4Department of Experimental Animal Facility, Daegu Catholic University Medical Center, Daegu 42472, Republic of Korea; 5Department of Radiological Sciences, School of Health Sciences, International University of Health and Welfare, Otawara 324-8501, Japan

**Keywords:** synchrotron radiation, early-stage lung cancer, virtual histology, differential diagnosis

## Abstract

The subtle deterioration in alveolar structures that occurs in the course of lung cancer development is often hard to detect using conventional imaging methods. We explored the application of X-ray methods using synchrotron radiation to reveal subtype-specific features with high clarity, including invasive fronts, necrotic foci, fibrotic bands, and deformation of alveolar microstructures, enabling non-destructive three-dimensional virtual histology. The resulting detail and contrast are comparable to those achieved with light microscopy, and the specimen is kept intact for standard evaluation. The purpose of this study is to investigate a rapid, non-destructive imaging method that complements pathology, helps clinicians distinguish between lung cancer subtypes, and improves confidence in early diagnosis. If clinically adopted, it will improve early diagnosis, guide treatment planning, and facilitate new research on how tumors spread along air spaces.

## 1. Introduction

Conventional absorption-based X-ray imaging techniques have advanced substantially in recent decades and play a central role in the diagnosis, staging, and follow-up of lung cancer. In particular, chest computed tomography (CT) enables rapid, high-resolution cross-sectional imaging of the thorax and is widely used in routine clinical practice [1,2,3].

Despite these technological advances, diagnostic confirmation of lung cancer still depends on histopathological examination, as conventional CT cannot provide sufficient three-dimensional morphological detail to enhance clinical insight [4,5]. A fundamental limitation of conventional CT lies in its dependence on X-ray attenuation contrast. Biological soft tissues, including lung parenchyma, are composed predominantly of low-atomic-number elements and therefore exhibit minimal differences in attenuation coefficients [6,7].

As a result, subtle microanatomical features—such as alveolar architecture, tumor–parenchyma interfaces, and subtype-specific growth patterns—cannot be adequately resolved. [7,8,9]. This limitation is particularly critical in early-stage lung cancer and in distinguishing histological subtypes, in which microstructural organization has important diagnostic and prognostic implications [10,11,12].

Phase-contrast imaging (PCI) partially overcomes these constraints by exploiting X-ray phase shifts rather than absorption differences to enhance soft-tissue contrast [7,13]. Synchrotron-based PCI, which benefits from high photon flux, spatial coherence, and monochromaticity, enables visualization of fine tissue microstructures that approach the level of detail observed in histopathology. However, many existing PCI approaches remain limited in their ability to consistently delineate complex alveolar microarchitectures and tumor-associated structural remodeling in three dimensions [11,14,15,16].

X-ray Dark-Field Imaging (XDFI) is a crystal analyzer-based phase-contrast modality that enhances sensitivity to small-angle X-ray scattering and refraction arising from microstructural heterogeneity. By combining dark-field detection with refraction-contrast imaging, XDFI Computed Tomography (CT) provides high-resolution, non-destructive three-dimensional visualization of soft tissues [17,18,19,20].

Previous studies have demonstrated this technology’s ability to depict microstructural features in organs such as the brain, breast, and eye [21,22,23,24,25,26], suggesting potential applicability to pulmonary tissues [27,28,29].

In this study, we visualized the three-dimensional morphology of four types of lung cancer—in normal, transitional, and malignant regions—using synchrotron XDFI CT. Our aim was to evaluate the clinical potential of synchrotron microtomography in differentiating microstructural features across a variety of histological subtypes of lung cancer. By correlating imaging findings with histopathological architecture, we assessed this technique’s diagnostic value as a non-destructive, high-resolution imaging modality for pulmonary oncology.

## 2. Materials and Methods

### 2.1. Tissue Preparation

Human lung cancer tissues were obtained from patients who underwent surgical resection for their malignancies at Korea University Anam Hospital. All patients provided written informed consent prior to tissue collection. The specimens were harvested following the completion of routine diagnostic procedures under the guidance of two pulmonary pathologists (J.H. Lee and Y. Kang). The study protocol was approved by the Institutional Review Board of Korea University Anam Hospital (IRB number: 2019AN0242). Detailed information regarding the specimens is provided in Table 1.

Four tumor specimens were chosen to represent distinct and clinically relevant histopathological contexts rather than to provide statistical representation. These included acinar-predominant adenocarcinoma, adenocarcinoma after concurrent chemoradiation therapy, keratinizing squamous cell carcinoma, and metastatic hepatocellular carcinoma in the lung. Specimens were sorted to explore whether XDFI CT can visualize subtype-associated microarchitectural features and treatment-related stromal remodeling across diverse pathological settings. Given the limited number of samples, this study was designed as a feasibility and exploratory investigation, and no statistical generalization was intended.

Routine pathological processing involved airway inflation and fixation in 10% neutral-buffered formalin, followed by gross examination, cancer lesion mapping, serial sectioning at 3 mm intervals, paraffin embedding, hematoxylin and eosin (H&E) staining, and immunohistochemistry. For synchrotron radiation (SR) imaging, selected specimens were embedded in 1% agarose gel and transported to the BL-14B beamline at the Photon Factory, High Energy Accelerator Research Organization (KEK), Tsukuba, Japan.

### 2.2. X-Ray Source and Experimental Setup

Synchrotron radiation XDFI CT image acquisition was performed at beamline BL-14B of the Photon Factory. The incident X-ray beam was generated by a 5-Tesla superconducting vertical wiggler installed in a 2.5-GeV storage ring and monochromatized using a double-crystal monochromator (TRINITY CO., LTD, Tokyo, Japan).

The imaging system consisted of the following components: (1) an asymmetrically cut silicon monochromator–collimator (AMC) to broaden the beam and reduce horizontal divergence; (2) a Laue-type angle analyzer (LAA) consisting of asymmetrically cut Si (111) crystal plates; (3) a scintillator-based X-ray detector coupled with high-sensitivity optics; and (4) a motorized sample rotation stage (Figure 1).

An incident X-ray energy of 19.8 keV was applied to balance phase-contrast sensitivity and penetration depth for human lung soft tissues. At this energy, the refraction angles generated by alveolar septa and tumor–stroma interfaces are sufficiently large to be detected by analyzer-based optics while minimizing absorption-related attenuation. Detailed optical physics of the analyzer-based configuration are provided in the Appendix A.

The LAA consisted of 166 μm thick Si (111) plates with a 5° asymmetric cut. This asymmetric Laue-case configuration increases angular dispersion and improves sensitivity to subtle electron density gradients by modulating the crystal’s acceptance angle. As a result, small refraction angles produced at alveolar septa and tumor–stroma boundaries are selectively filtered and amplified, thereby enhancing phase sensitivity and soft-tissue contrast.

### 2.3. Imaging Parameters, Acquisition, and Reconstruction

Specimens were mounted on a motorized stage within a lead-shielded hutch. Each sample was placed in a 2 cm diameter acrylic cylinder containing 1% agarose to prevent motion during scanning.

Key imaging and reconstruction parameters, including voxel size, effective spatial resolution, field of view, total acquisition time, and reconstruction settings, are summarized in Table 2. The X-ray detector employed a camera with a physical pixel size of 2.75 μm; however, the effective spatial resolution of the entire imaging system, including the synchrotron source properties, silicon single-crystal optics, the scintillator in the detector, and geometric blurring, was approximately 10 μm. Each tomographic scan required approximately 3 h, with 2500 projections acquired over a full 360° rotation.

Three-dimensional reconstructions were performed using Amira-Avizo software (version 2020.3, Thermo Fisher Scientific, Burlington, MA, USA). Volume rendering and visualization were conducted using GPU-accelerated processing to assess fine microstructural features. Detailed imaging workflows for individual specimens are provided in Appendix B (Figure A1, Figure A2, Figure A3 and Figure A4).

### 2.4. Comparison with Histopathology

After synchrotron imaging, all specimens were returned for standard paraffin embedding and histological evaluation. Serial H&E-stained sections were reviewed by board-certified pathologists. To ensure spatial consistency between XDFI CT images and histology, the regions of interest selected for high-resolution imaging were matched to histological sections based on gross specimen orientation and anatomical landmarks.

XDFI CT images and histopathology were qualitatively compared to validate morphological correlates, including tumor–normal tissue interfaces, stromal remodeling, necrosis, and keratinization. Full-field synchrotron images and corresponding histological slides, with region-of-interest localization indicated by inset boxes, are provided in Appendix B. A comparative overview of light microscopy, absorption-based micro-CT, and synchrotron XDFI CT is provided in Appendix A.

## 3. Results

Representative regions of Interest (ROIs) were magnified and compared with pathologic findings for each specimen to highlight the characteristic features of each malignancy.

### 3.1. Comparison of Images from Primary Adenocarcinoma, Acinar-Predominant

Specimen 1 was obtained from a surgically resected primary lung adenocarcinoma with an acinar-predominant pattern (Figure 2). Histopathological examination demonstrated invasive acinar structures composed of well-formed glandular lumina arranged back-to-back within desmoplastic stroma, with focal lepidic growth at the tumor–parenchyma interface. Limited areas suspicious for solid differentiation were also identified beneath the pleura.

On the corresponding XDFI CT image, lepidic components were visualized as mildly thickened alveolar septa with preservation of the underlying alveolar framework, whereas invasive acinar regions appeared as irregular, thickened microarchitectural bands associated with distortion and partial loss of normal alveolar geometry. The surrounding alveolar septa and microvascular structures were clearly delineated and traceable over millimeter-scale distances, reflecting strong refraction contrast at air–tissue interfaces.

Focal micropapillary components (two navy arrows in Figure 2b) and histologically suspected solid patterns were not distinctly identified in the XDFI image, indicating the limit of the current spatial resolution for detecting fine architectural features at the cellular or subalveolar scale. Full-field synchrotron images and images of the corresponding histological sections for this specimen are provided in Appendix A.

### 3.2. Images from Primary Adenocarcinoma Resected After CCRT

Specimen 2 was obtained from a primary lung adenocarcinoma with an acinar-predominant pattern that was resected after concurrent chemoradiotherapy (CCRT) (Figure 3). Histopathological examination demonstrated extensive treatment-related remodeling characterized by broad areas of necrotic and hyalinized desmoplastic stroma with scattered residual acinar glands. The tumor–lung interface showed prominent fibrosis and hemorrhagic change, consistent with post-therapeutic regression, while the peripheral lung parenchyma showed preserved normal alveolar architecture.

The XDFI CT images showed high-attenuation, low-porosity bands composed of multilayered microstructural plates, which correlated with treatment-induced necrotic and fibrotic regions found via H&E staining. Linear lamellar densities and coarse trabeculations within these regions corresponded to stromal hyalinization and scar formation observed histologically. Residual viable tumor components were visualized as focal interruptions within the fibrotic plate and were spatially concordant with scattered acinar glands on histopathology.

Areas of hemorrhagic congestion showed increased attenuation with reduced phase-contrast heterogeneity within alveolar spaces, consistent with alveolar hemorrhage and radiation-induced vascular injury. Full-field synchrotron images and corresponding histological sections for this specimen are provided in Appendix A.

### 3.3. Images from Squamous Cell Carcinoma (Figure 4)

Specimen 3 was an invasive keratinizing-type squamous cell carcinoma of the lung (Figure 4). Histopathological examination showed invasive nests and sheets of malignant squamous cells embedded within desmoplastic stroma, with prominent keratinization and central necrosis. A circumferential fibrotic rim sharply demarcated the tumor from the adjacent aerated lung parenchyma, and focal hemorrhagic and congestive changes were observed at the tumor periphery.

**Figure 4 cancers-18-00082-f004:**
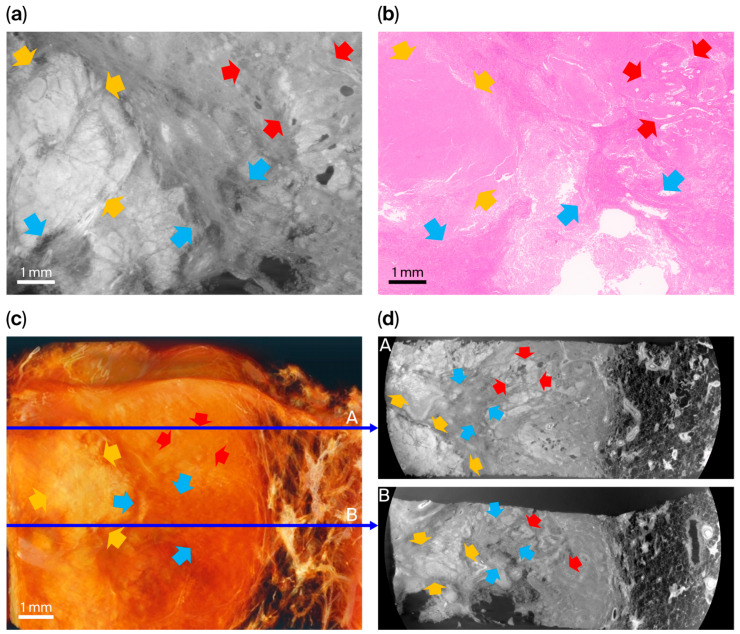
Comparison between XDFI CT and histopathologic findings for specimen 3 (primary lung squamous cell carcinoma, keratinizing type). (**a**) Representative sagittal section from XDFI CT. (**b**) Corresponding hematoxylin and eosin-stained histologic section from the same region. (**c**) Three-dimensional volume-rendered reconstruction of the specimen. (**d**) Axial views extracted from the reconstructed volume at different depths (1/3; A, and 2/3; B, indicated by thin blue arrows). Keratinizing and necrotic tumor foci (yellow arrows) within desmoplastic stroma showed broad eosinophilic keratinization with central necrosis and surrounding fibrotic reaction on histology, indicating regressive changes in keratinizing nests. On synchrotron images, these areas appeared as high-attenuation regions with complete loss of normal alveolar architecture. Viable invasive tumor components (cyan arrows) consisted of malignant squamous cells forming irregular sheets and nests infiltrating desmoplastic stroma, consistent with keratinizing squamous differentiation. Hemorrhagic and congestive zones (red arrows) demonstrated vascular dilatation with extravasated erythrocytes and peritumoral edema on histology; on XDFI CT images, these regions corresponded to fluid-filled alveolar spaces with reduced air–tissue phase contrast.

In the XDFI CT images, the tumor appeared as a compact, high-attenuation, low-porosity domain in sharp contrast to the surrounding low-attenuation alveolar structures. Keratinizing and necrotic tumor regions showed complete loss of normal alveolar architecture and markedly increased attenuation, whereas viable invasive tumor components appeared as dense internal structures with irregular microarchitectural organization. A pronounced refractive gradient at the tumor–parenchyma interface corresponded to the peripheral fibrotic rim identified histologically.

Regions of hemorrhagic congestion presented as fluid-filled alveolar spaces with reduced air–tissue phase contrast, consistent with vascular dilatation and extravasation observed on histopathology. Full-field synchrotron images and the corresponding histological sections for this specimen are provided in Appendix A.

### 3.4. Images of Metastatic Carcinoma from Hepatic Cellular Carcinoma

Specimen 4 was a metastatic carcinoma in the lung originating from hepatocellular carcinoma (HCC) (Figure 5). Histopathological examination demonstrated a well-circumscribed, expansile tumor composed of polygonal malignant cells arranged in trabecular plates, which is characteristic of hepatocellular differentiation. The lesion was sharply demarcated from the surrounding lung parenchyma by a thin fibrotic stromal rim, and broad degenerative clefts dissected the tumor into lobulated compartments.

The metastatic lesion appeared as a homogeneous, compact, high-attenuation, low-porosity domain distinct from the surrounding low-attenuation alveolar architecture in XDFI CT image. The tumor–parenchyma interface was visualized as a continuous, high-visibility phase fringe corresponding to the collagen-enriched peripheral stromal rim identified histologically.

Multiple linear and fissure-like low-transmission defects were observed within the tumor mass. These were spatially concordant with degenerative clefts seen on histopathology. These defects exhibited prominent edge enhancement due to strong refractive index gradients, while the internally compact trabecular plates demonstrated reduced dark-field scattering, consistent with densely packed tumor cell architecture. Full-field synchrotron images and corresponding histological sections for this specimen are provided in Appendix A.

### 3.5. Three-Dimensional Segmentation Reconstruction Images of All Specimens

Figure 6 shows parts of the three-dimensional reconstruction images of each specimen. Spiculated adenocarcinoma tumor spikes invading loose, weave-like normal lung parenchyma were noted, and a conspicuous, clearly demarcated round border was visible in the metastatic hepatocellular carcinoma. Irregularly shaped necrotic hyalinized areas appear relatively bright compared to viable tumorous lesion in both post-CCRT adenocarcinoma and squamous cell carcinoma, which can be distinguished by the homogenous dense compartment of metastatic cancer.

## 4. Discussion

In this study, we successfully reproduced the histologic characteristics of several lung cancer subtypes at a micrometer-scale resolution, including precise delineation of tumor–normal tissue interfaces and visualization of fibrotic stromal remodeling, keratinization, hyalinization, and the architectural changes associated with neoadjuvant chemoradiation therapy.

Synchrotron radiation imaging with XDFI optics accurately reflected the border between condensed tumor and normal alveolar structures, enabling clear structural distinction across different histological entities, including adenocarcinoma with and without neoadjuvant chemotherapy, squamous cell carcinoma, and metastatic tumors. [30,31].

Despite the high microstructural fidelity achieved with synchrotron radiation XDFI, several important limitations must be acknowledged. Although spatial resolution has been continuously improved in analyzer-based refraction-contrast tomography [19,21,22,32], the current XDFI CT configuration remains insufficient to reliably resolve critical microscale architectural features, such as focal micropapillary components and tumor spread through air spaces (STAS), which are known to be decisive for prognostic stratification in lung adenocarcinoma. These resolution constraints also limit the accurate differentiation of certain histologic subtypes, including papillary, solid, cribriform, and mucinous patterns [33].

Moreover, while synchrotron radiation-based micro-CT techniques have demonstrated strong regional correspondence with histologic architecture, current imaging signatures remain insufficiently specific for independent diagnostic interpretation without correlative pathology [11,29,34,35]. Similarly, although necrosis and hyalinization generate prominent contrast differences on XDFI CT, these features cannot yet be translated into quantitative surrogates for viable tumor burden in the absence of validated objective metrics [36].

Certain considerations should be taken into account when interpreting the outcomes of this study. First and foremost, the sample size was small (*n* = 4); the findings should therefore be interpreted as preliminary and exploratory rather than representative of all lung cancer subtypes. The restricted field of view of the BL-14B imaging geometry also limits whole-organ visualization and prevents comprehensive mapping of tumor architecture in a broader anatomic context. In addition, the relatively long acquisition times required for analyzer-based refraction-contrast imaging currently impose practical constraints on scalability. Moreover, this modality is inherently limited to ex vivo application; the radiation dose, beamline infrastructure, and optical configuration preclude its direct implementation in in vivo settings.

Another essential limitation of this study is the absence of quantitative analysis. Although qualitative correspondence between synchrotron refraction-contrast imaging and histopathology provides meaningful preliminary insight, reliable voxel-to-slide co-registration could not be achieved. Lung tissue is particularly susceptible to fixation-related deformation and anisotropic shrinkage, which complicate precise spatial matching between synchrotron datasets and histological sections. Consequently, quantitative microarchitectural validation could not be incorporated into this investigation.

We consider this limitation a critical area for further development and are actively pursuing follow-up studies aimed at establishing reproducible quantitative indices that integrate refraction-contrast signatures with pathological ground truth. These efforts include the development of metrics related to alveolar wall morphology, phase-edge sharpness, acinar porosity, and treatment-related stromal alterations, which are expected to form the methodological foundation for future extensions of this work.

These limitations delineate important methodological requirements for advancing three-dimensional virtual histology. In particular, there is a need to formalize quantitative imaging features that integrate attenuation, phase-edge amplitude, and subvoxel-scattering proxies with robust voxel-to-slide registration supported by stereologic validation and pathological ground truth [37,38,39]. Machine learning approaches incorporating these image-derived features—especially for non-collagenated and heterogeneous tissue architectures—may further improve histological subtype discrimination and reproducibility [35,40,41]. In parallel, protocol refinements, including phase-retrieval calibration, optimized energy selection, and improved detector deconvolution, are expected to enhance sensitivity to subtle refractive microstructures [42,43,44].

To contextualize the abilities of the Photon Factory (PF) within the international synchrotron landscape, it is helpful to consider developments at leading facilities. The European Synchrotron Radiation Facility (ESRF), operating an Extremely Brilliant Source, achieves unparalleled photon flux and spatial coherence, enabling phase-contrast and microbeam imaging at submicron resolution and supporting functional in situ experiments in soft tissues [45,46,47,48,49]. The ESRF is currently at the forefront of biomedical synchrotron imaging, particularly in nanoscale and multiscale applications [50].

The SYRMEP beamline at Elettra Synchrotrone Trieste prioritizes versatile, high-resolution microtomography for biomedical and biomaterial specimens, with flexible energy selection and pixel sizes down to several micrometers at both laboratory and synchrotron scales [47,51]. Ongoing developments aim to bridge laboratory micro-CT and synchrotron performance through multiscale imaging strategies, supported by continued upgrades to optics and detectors that expand usable pixel-size ranges while maintaining field-of-view and workflow efficiency [52,53,54].

The Imaging and Medical Beamline (IMBL) at the Australian Synchrotron is specifically engineered for ultra-large-field-of-view clinically oriented imaging, combining propagation-based phase-contrast and absorption modalities in the 25–120 keV range to visualize organ-scale structure and function, including whole-lung and vascular networks [55,56]. Recent work has emphasized rigorous beam characterization, protocol standardization, and translational pipelines extending from preclinical studies to clinical implementation, enabled by long beamline geometry and high-power insertion devices [15,35,57,58]. Future developments are directed toward integrated, low-dose, high-sensitivity platforms for diagnosis, therapy planning, and treatment response assessment [58,59,60,61,62,63].

Against this backdrop, recent advances in synchrotron-based biomedical imaging increasingly emphasize multiscale and multilayer three-dimensional visualization supported by higher source brilliance and extended beamline infrastructures. Notably, the ESRF operates a fourth-generation synchrotron source, whereas SYRMEP and IMBL are based on advanced third-generation accelerators, with the ESRF and IMBL equipped with dedicated medical long beamlines and SYRMEP actively pursuing long-beamline upgrades.

In this context, the Photon Factory’s approach demonstrates particular strength in the high-resolution imaging of low-attenuation tissues such as the human lung, despite its relatively limited field of view. Although this constraint currently precludes whole-organ imaging and represents a barrier to direct clinical translation, this study shows that carefully optimized optical design and reconstruction strategies can partially compensate for the infrastructure limitations inherent to a second-generation synchrotron facility.

Beyond facility-specific considerations, meaningful distinctions among phase-contrast imaging approaches arise from their underlying contrast mechanisms and optical configurations. Propagation-based phase-contrast imaging is well-suited to large fields of view and multiscale studies but relies strongly on propagation distance and phase-retrieval assumptions. Grating-based interferometric techniques provide quantitative phase information but face limitations in spatial resolution, dose efficiency, and scalability for thick or highly heterogeneous specimens. In contrast, crystal analyzer-based X-ray dark-field imaging (XDFI), as employed in this study, offers exceptional sensitivity to subtle refraction-angle variations at air–tissue and tissue–tissue interfaces, enabling high-contrast visualization of alveolar microarchitecture and fibrotic remodeling without exogenous contrast agents.

Our study shows that analyzer-based XDFI, when combined with optimized acquisition and reconstruction strategies, can yield biologically meaningful three-dimensional virtual histology even within the constraints of a second-generation synchrotron environment. For broader applicability and potential translational relevance, future efforts should focus on improving spatial resolution, expanding the effective field of view while maintaining micrometer-scale detail, and accelerating acquisition. In parallel, integrating quantitative feature extraction and machine learning-based classification—incorporating attenuation, phase-edge amplitude, porosity spectra, and dark-field surrogates—will be essential to improving robustness and generalizability across histological subtypes and post-treatment states.

## 5. Conclusions

This study shows the applicability of synchrotron radiation XDFI CT, which enables high-resolution, non-destructive three-dimensional visualization of lung cancer microarchitecture and provides meaningful structural correspondence with histopathological findings. By revealing subtype-specific growth patterns, stromal remodeling, and treatment-related architectural changes, XDFI CT complements conventional pathology with volumetric context and spatial continuity.

While the current system is limited by its spatial resolution, field of view, and lack of quantitative validation, the results establish a technical and conceptual foundation for the further development of three-dimensional virtual histology in lung cancer research. Future efforts focusing on resolution enhancement, quantitative analysis, and multiscale imaging will be necessary to advance the translational relevance of this approach.

## 6. Patents

The specimens used in this study were harvested from patients who underwent surgical resection for their disease. Donations were obtained with written informed content from the patients before enrollment. The patient characteristics are described in Table 1.

## Figures and Tables

**Figure 1 cancers-18-00082-f001:**
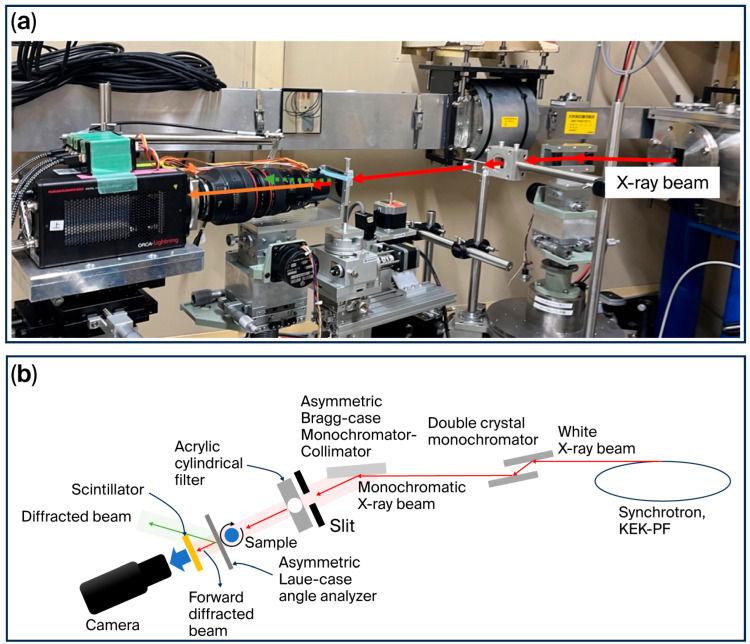
Refraction-contrast CT imaging system constructed at KEK-PF BL-14B. (**a**) Photograph of the overall imaging setup. (**b**) Schematic diagram of the imaging (top view).

**Figure 2 cancers-18-00082-f002:**
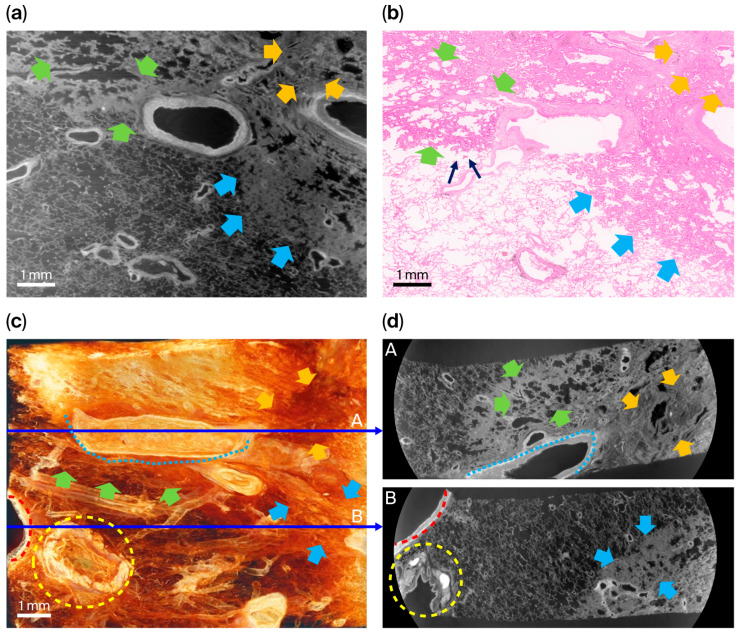
Comparison between synchrotron radiation XDFI CT image and histopathologic findings in specimen 1 (primary lung adenocarcinoma, acinar-predominant). (**a**) Representative sagittal section XDFI CT. (**b**) H&E-stained histologic section from the same region. (**c**) Three-dimensional volume-rendered reconstruction of the specimen. (**d**) Axial views extracted from the reconstructed volume at different depths (1/3; A, and 2/3; B, indicated by thin blue arrows). On histology, lepidic growth (green arrows) was characterized by neoplastic pneumocytes lining pre-existing alveolar septa without stromal invasion, while acinar components (cyan arrows) showed invasive glandular proliferation, forming irregular acini within desmoplastic stroma. Areas suspicious for solid differentiation (yellow arrows) exhibited loss of glandular or alveolar architecture. Sporadic micropapillary tufts (navy arrows) were identified only in histopathologic examination. Corresponding XDFI CT images demonstrate preserved alveolar frameworks and microvascular structures in lepidic regions, whereas invasive acinar areas showed thickened and distorted microarchitectural bands. Major vascular structures (dashed cyan lines for venules; dashed red lines for arterioles) and cartilaginous bronchioles (dashed yellow lines) were identifiable and traceable in three dimensions.

**Figure 3 cancers-18-00082-f003:**
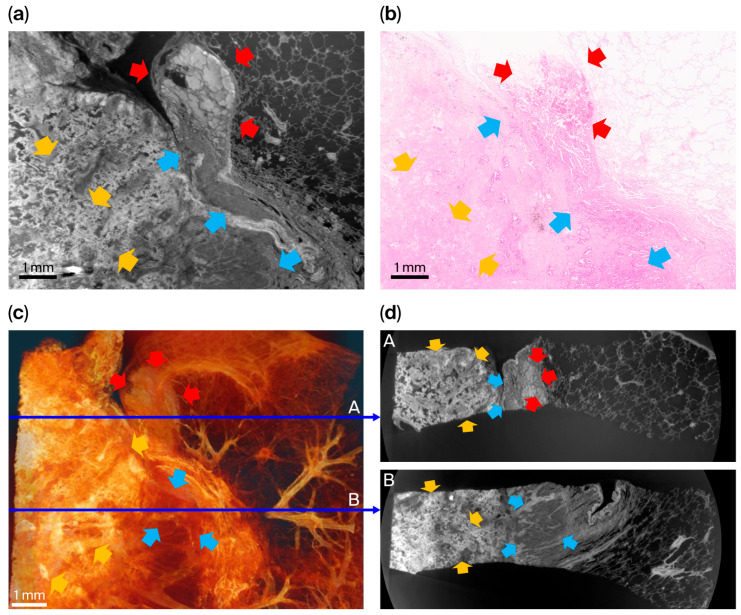
Comparison of XDFI CT and histopathologic findings in specimen 2 (primary lung adenocarcinoma, acinar-predominant, post-CCRT). (**a**) Representative sagittal section from XDFI CT. (**b**) Corresponding H&E-stained histologic section from the same region. (**c**) Three-dimensional volume-rendered reconstruction of the specimen. (**d**) Axial views extracted from the reconstructed volume at different depths (1/3; A, and 2/3; B, indicated by thin blue arrows). On histology, treatment-related necrotic desmoplastic stroma (red arrows) showed extensive fibrosis and tissue replacement following chemoradiotherapy. In the XDFI CT images, these regions appeared as compact, multilayered band-like structures with low porosity and disrupted microstructural density. Residual viable tumor components (cyan arrows) consisted of adenocarcinoma cells forming irregular acinar or gland-like nests embedded within partially fibrotic stroma; these correspond to focal interruptions within the fibrotic plate on SR images. Hemorrhagic congestion zones (red arrows) demonstrated increased attenuation and diminished phase-contrast heterogeneity within alveolar spaces, reflecting alveolar hemorrhage and vascular congestion secondary to radiation-induced vascular injury.

**Figure 5 cancers-18-00082-f005:**
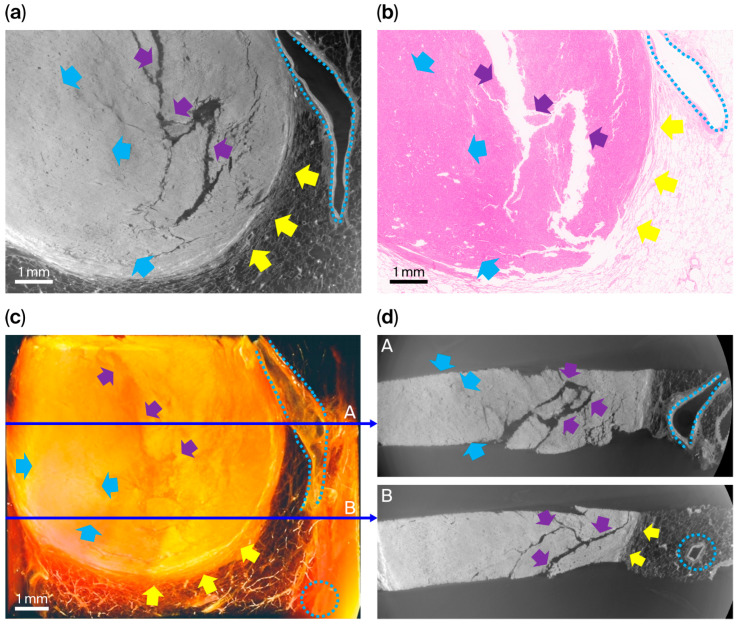
Comparison of XDFI CT and histopathologic findings for specimen 4 (pulmonary metastasis from hepatocellular carcinoma). (**a**) Representative sagittal section from synchrotron-based XDFI tomography. (**b**) Corresponding hematoxylin and eosin-stained histologic section from the same region. (**c**) Three-dimensional volume-rendered reconstruction of the specimen. (**d**) Axial views extracted from the reconstructed volume at different depths (1/3; A, and 2/3; B, indicated by thin blue arrows). Viable metastatic tumor components (cyan arrows) exhibited a trabecular growth pattern formed by polygonal tumor cells arranged in interlacing cords or plates, characteristic of hepatocellular carcinoma. In synchrotron images, these regions appeared as compact, high-attenuation domains with low internal porosity. Intra-tumoral clefts and fissure-like defects (purple arrows), caused by condensation and retraction of tumor cell nests, corresponded to necrotic or fibrotic clefts on histology and are visualized on SR images as linear low-transmission structures with prominent edge enhancement. A well-defined peripheral stromal rim (bright yellow arrows) sharply demarcated the tumor from surrounding aerated alveolar parenchyma and appeared on SR images as a continuous high-density boundary reflecting collagen-enriched tissue. Major vascular structures were identifiable (mostly veins, demarcated with cyan dotted lines) and traceable.

**Figure 6 cancers-18-00082-f006:**
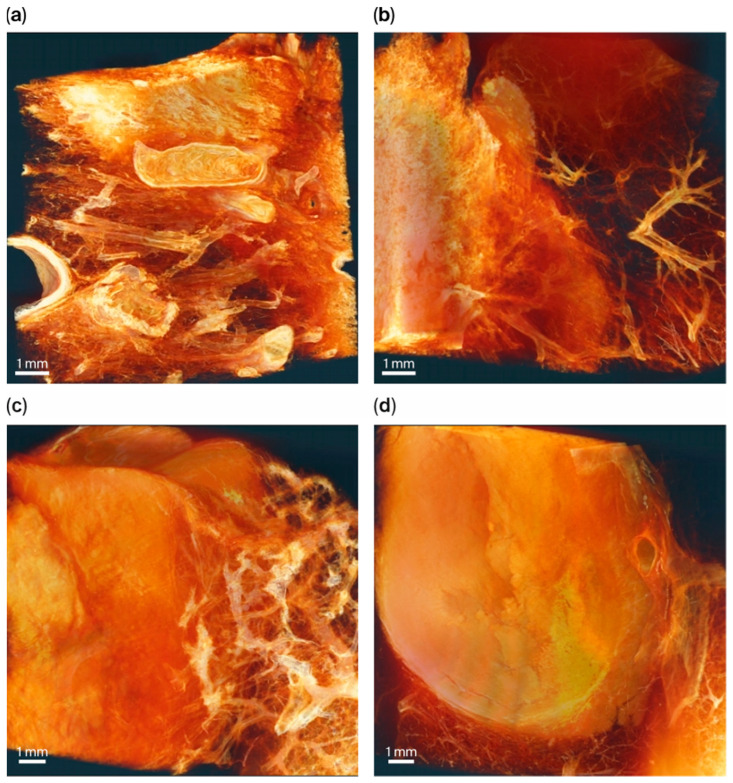
Segmentation 3D images of samples. (**a**) Specimen 1; (**b**) specimen 2; (**c**) specimen 3; (**d**) specimen 4. Full video images are provided in the Appendix A (Appendix A for specimen 1, Appendix A for 2, Appendix A for 3, and Appendix A for 4).

**Table 1 cancers-18-00082-t001:** Summary of clinical–pathologic information of lung specimens (*n* = 4).

Case No.	Age	Gender	Tumor Location	Pathologic Stage	Histopathological Description
#1	84	Woman	RLL ^1^	Primary lung cancer (pT2N0M0)	Adenocarcinoma, acinar-predominant
#2	65	Woman	RLL	Primary lung cancer (pT2N0M0)	Adenocarcinoma, acinar-predominant; post CCRT ^2^
#3	77	Man	RLL	Primary lung cancer (pT2N0M0)	Squamous cell carcinoma, keratinizing
#4	63	Woman	RLL	Metastatic lung cancer from HCC	Metastatic hepatocellular carcinoma

^1^ RLL; right lower lobe, ^2^ CCRT; concurrent chemoradiation therapy.

**Table 2 cancers-18-00082-t002:** Comparison of experimental settings.

Variable		Condition
Incident X-ray		
	X-ray energy	Monochromatic 19.8 keV
	Diffraction plane of double-crystal (MC ^1^)	Symmetric Bragg-case Si (111) 23 ^H^ × 21 ^V^ mm^2^
	Number of photons	Approximately 108 photons/mm^2^/s
	Measurement time for sample	3 h
AMC ^2^		
	Diffraction plane	Asymmetric Bragg-case Si (111)
	Thickness	20.5 mm
	Area	124.8 ^H^ × 42.8 ^V^ mm^2^
	Asymmetric angle	5.4°
LAA ^3^		
	Diffraction plane	Asymmetric Bragg-case Si (111)
	Thickness	166 μm
	Area	55 ^H^ × 50 ^V^ mm^2^
	Asymmetric angle	5.0°
Sample stage		
	Step angle	0.144°
	Rotation angle	360°
	Number of projections	2500
X-ray camera		data
	Optic camera	ORCA-Lightning digital CMOS ^5^ camera
		Hamamatsu Photonics K. K
	FOV ^4^	14.6 ^H^’ × 12.7 ^V^ mm^2^
	Pixel size	2.75 μm
	X-ray scintillator	LuAG: Ce ^6^ thickness: 100 μm
	Lens optics	85 mm, F1.2, Canon inc. (Tokyo, Japan)

^1^ MC—monochromator collimator, ^2^ AMC—asymmetric Bragg-case monochromator collimator, ^3^ LAA—Laue angular analyzer ^4^ FOV—field of view, ^5^ CMOS—complementary metal-oxide semiconductor, ^6^ LuAG: Ce—lutetium aluminum g00arnet doped with cerium.

## Data Availability

The authors confirm that the data supporting the findings of this study are available in the Appendix A.

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
