# Peer review of "Potential of Higher Resolution Synchrotron Radiation Tomography Using Crystal Analyzer-Based Imaging Techniques for Differential Diagnosis of Human Lung Cancers"

_cancers, 2025, doi:10.3390/cancers18010082_

Round 1
Reviewer 1 Report
Comments and Suggestions for Authors
This manuscript evaluates the capability of synchrotron radiation–based refraction-contrast XDFI microtomography to reproduce histological microstructures in four human lung cancer subtypes. The authors demonstrate clear visualization of tumor margins, fibrotic remodeling, necrosis, keratinization, vascular networks, and post-CCRT changes, with strong correlation to H&E sections. The technique shows promise for non-destructive 3D virtual histology and may support subtype differentiation and treatment-response assessment.
This study is well described. However, there are some points need to be addressed:
- Please enhance the methodological clarity. Additional explanation is needed regarding the choice of X-ray energy, the optical configuration of the Laue angle analyzer, and how these parameters influence spatial resolution and phase sensitivity.
- Please strengthen the quantitative validation. The current comparisons are largely qualitative. Consider including metrics such as signal intensity profiles, porosity maps, or refractive-index contrast measurements, or clearly state this as a limitation.
- Please discuss limitations more explicitly. The manuscript should acknowledge constraints such as small sample size (n=4), limited FOV, long scanning times, and the lack of in-vivo applicability.
- Please provide a more structured comparison with other synchrotron-based phase-contrast modalities. The discussion focuses heavily on facility descriptions (ESRF, SYRMEP, IMBL). Comparisons should shift toward technical distinctions and practical advantages/limitations of XDFI versus alternative phase-contrast approaches.
- Please standardize arrow colors, labels, and scale bars across all figures to improve readability.
- Please improve the resolution of Supplementary Figures S2–S5, where some labels are difficult to interpret.
- Please ensure consistent terminology (e.g., “SR-uCT,” “XDFI tomography,” “refraction-contrast imaging”).
- Please add inset boxes on histology figures indicating precisely which region corresponds to the SR slice.
- Please review formatting and units (pixel size, thickness, beam parameters) for consistency.
Please revise the manuscript carefully for English clarity, grammar, and readability. Below are specific areas that would benefit from improvement:
- Many sentences combine multiple ideas without appropriate punctuation, making the meaning difficult to parse. For example, in the Introduction and Discussion, several sentences exceed 40–50 words and include multiple subordinate clauses.
Please break these into shorter, more direct statements. - Certain concepts, such as contrast mechanisms, alveolar microstructures, and the advantages of synchrotron radiation, are repeated multiple times with slightly different wording. Please consolidate repetitive descriptions to enhance readability and avoid redundancy.
- Some sections switch between past and present tense when describing the same experiment or result. Please standardize to past tense for methods and results, and reserve present tense for interpretation or established knowledge.
- Terms such as “enhanced signals,” “grayish materials,” “porosity interruptions,” and “bright lines” need clearer definition or more standardized scientific wording. Please clarify these expressions to ensure precise communication. Examples include: “multiple sheet-like, mildly thickened alveolar walls”. “compact multi-layered band structures with rare porosity”. “low-attenuated dots-like structures”. These phrases are understandable but not idiomatic English. Please revise for scientific clarity and more natural expression.
6. Descriptions of imaging findings often lack parallel structure. When comparing SR images to histology, the syntax becomes uneven and difficult to follow. Please use consistent sentence structure when describing paired findings (e.g., SR → histology → interpretation).
7. Some transitions between sentences and paragraphs are abrupt. For example, in the Results section, transitions between tumor subtypes are sudden. Please add short linking sentences to guide the reader from one idea to the next.
8. Some grammatical corrections are needed: Examples include: “identified as multiple sheet-like…” (should be “were identified as…”). “interruption within the fibrotic plate are spatially concordant…” (subject–verb agreement issue). “resemble to the fibrotic tumor bed” (should be “resemble the…”). Please revise these grammatical inconsistencies.
9. Please correct the followings: missing spaces (“3Dreconstruction”), inconsistent hyphenation (“tumor-normal parenchymal margin”). Please correct typographical inconsistencies throughout.
10. Improve figure captions for readability. Many captions contain long blocks of text with multiple findings in a single sentence. Please break them into shorter, structured sentences (e.g., describing normal, malignant, transitional zones separately).
Author Response
|
The authors would like to thank Reviewers for careful review of our manuscript and providing us with their comments and suggestion to improve the quality of the manuscript. The following responses have been prepared to address all of the editor’s comments in a point –by-point fashion. |
Comments and Suggestions for Authors
This manuscript evaluates the capability of synchrotron radiation–based refraction-contrast XDFI microtomography to reproduce histological microstructures in four human lung cancer subtypes. The authors demonstrate clear visualization of tumor margins, fibrotic remodeling, necrosis, keratinization, vascular networks, and post-CCRT changes, with strong correlation to H&E sections. The technique shows promise for non-destructive 3D virtual histology and may support subtype differentiation and treatment-response assessment.
This study is well described. However, there are some points need to be addressed:
Point-by-point response to Comments and Suggestions for Authors
Comment 1
Please enhance the methodological clarity. Additional explanation is needed regarding the choice of X-ray energy, the optical configuration of the Laue angle analyzer, and how these parameters influence spatial resolution and phase sensitivity.
Answer 1
I and my coauthors sincerely appreciate for the reviewer’s thoughtful suggestion regarding the need for improving methodological clarification. We have added sentences for improving methodological clearance, regarding with (1) why we chose the incident X-ray energy of 19.8keV, (2) the optical configuration of Laue-Angle Analyzer (LAA), and (3) the way of these parameters affecting spatial resolution, phase sensitivity, and overall imaging performances, according to the reviewer’s advices.
The incident X-ray energy of 19.8keV was selected because we thought that it provides an optimal balance between phase-contrast sensitivity and beam transmission for human lung tissues. At this energy, the refractive index decrement (δ) of low-atomic-number soft tissues yields sufficiently large refraction angles for analyzer-based phase detection, while minimizing absorption-related signal loss. This energy also corresponds to the most stable operating regime of the BL-14B beamline, ensuring a highly coherent and monochromatic beam suited for dark-field imaging optics.
Also, we would like to add the sentences explaining clarification on the asymmetric Laue-case configuration of the angle analyzer. The thin Si (111) plates (166μm) with a 5° asymmetry angle provide angular filtering of refracted X-rays with high sensitivity to subtle electron-density gradients. The asymmetric cut enhances angular dispersion (b-factor modulation), improving detection of minute refraction angles arising from alveolar septa and stromal microstructures.
Finally, we would like to add sentences for describing how these parameters jointly determine the effective spatial resolution (~10μm). While the detector pixel size is 5.5μm, the ultimate resolution is shaped by analyzer bandwidth, beam divergence, sample-to-analyzer geometry, and the angular acceptance of the LAA. By refining this description in the revised manuscript, we aim to provide greater transparency and reproducibility.
Response 1
We would like to insert sentences to supplement the insufficient explanation of the Materials and Methods section.
In 2.2. X-ray source and Experimental setup of part 2 Materials and Methods section. We would like to insert two additional paragraphs describing (1) why we chose the incident X-ray energy of 19.8keV, (2) the optical configuration of Laue-Angle Analyzer (LAA). Changes were marked with yellowish color in the text.
An incident X-ray energy of 19.8 keV was applied to balance phase-contrast sensitivity and penetration depth for human lung soft tissues. At this energy, re-fraction angles generated by alveolar septa and tumor–stroma interfaces are sufficiently large to be detected by analyzer-based optics while minimizing absorption-related attenuation. Detailed optical physics of the analyzer-based configuration are provided in the Supplementary Materials (Figure S1).
The LAA consisted of 166-μm-thick Si (111) plates with a 5° asymmetric cut. This asymmetric Laue-case configuration increases angular dispersion and im-proves sensitivity to subtle electron-density gradients by modulating the crystal’s acceptance angle. As a result, small refraction angles produced at alveolar septa and tumor–stroma boundaries are selectively filtered and amplified, thereby enhancing phase sensitivity and soft-tissue contrast.
In 2.3. Imaging acquisition and Reconstruction of part 2 Materials and Methods section. We would like to insert two additional paragraphs describing (3) the way of these parameters affecting spatial resolution, phase sensitivity, and overall imaging performances. Newly inserted paragraphs were highlighted with yellowish color in the text.
The X-ray detector employed a camera with a physical pixel size of 2.75μm; however, the effective spatial resolution of the entire imaging system, including the synchrotron source properties, silicon single-crystal optics, scintillator in the detector, and geometric blurring, was approximately 10μm
Comment 2
Please strengthen the quantitative validation. The current comparisons are largely qualitative. Consider including metrics such as signal intensity profiles, porosity maps, or refractive-index contrast measurements, or clearly state this as a limitation.
Answer 2.
We are truly grateful for the reviewer’s precise comment. We totally agree that the quantitative analysis including metrics such as signal intensity profiles, porosity maps, or refractive-index contrast measurements should be mentioned, and would like to deeply apologize that we have not described any of quantitative metrics.
In this study, we focused on the qualitative comparison between refraction-contrast images using synchrotron radiation and pathologic examinations. Quantitative metrics—such as signal intensity profiles, porosity distributions, or refractive-index contrast measurements—were not incorporated because voxel-level registration between SR datasets and histologic sections remains technically challenging in heterogeneous lung specimens. The intrinsic tissue deformation that occurs during formalin fixation, paraffin embedding, and sectioning prevents accurate one-to-one correspondence with the intact three-dimensional SR volume, thereby limiting the feasibility of deriving reliable quantitative microstructural measurements within the scope of this feasibility study.
We are thoroughly aware that quantitative validation will be essential those kinds of studies and would like to compensate this weakness in the subsequent research work. We have therefore revised the manuscript to explicitly discuss this limitation and outline a future methodological framework that includes voxel-to-slide co-registration, automated segmentation pipelines, and quantitative extraction of structural parameters such as alveolar wall thickness, acinar porosity, phase-edge amplitude, and necrotic burden. These developments will allow more robust statistical assessment and enhance the diagnostic utility of SR-based virtual histology in the following our studies.
Response 2.
We would like to add paragraphs describing that the absence of quantitative analysis was one of main limitation with our current study in the Discussion section, according to the reviewer had pointed out. Corrections were highlighted with yellowish color in the text.
Another essential limitation of the present study is the absence of quantitative analysis. Although qualitative correspondence between synchrotron refraction-contrast imaging and histopathology provides meaningful preliminary insight, reliable voxel-to-slide co-registration could not be achieved. Lung tissue is particularly susceptible to fixation-related deformation and anisotropic shrinkage, which complicate precise spatial matching between synchrotron datasets and histological sections. Consequently, quantitative microarchitectural validation could not be incorporated into the current investigation.
We consider this limitation as a critical area for further development and are actively pursuing follow-up studies aimed at establishing reproducible quantitative indices that integrate refraction-contrast signatures with pathological ground truth. These efforts include the development of metrics related to alveolar wall morphology, phase-edge sharpness, acinar porosity, and treatment-related stromal alterations, which are expected to form the methodological foundation for future extensions of this work.
Comment 3.
Please discuss limitations more explicitly. The manuscript should acknowledge constraints such as small sample size (n=4), limited FOV, long scanning times, and the lack of in-vivo applicability.
Answer 3
We sincerely thank the reviewer for this valuable advice. Indeed, we are fully aware that the limitation induced by a small study population should be mentioned more explicitly to help readers appropriately interpret the scope and implications of our findings. In response, we have revised the Discussion section to provide a clearer and more comprehensive description of the constraints associated with our current work.
In the revised discussion, we would like to explicitly acknowledge that the sample size is small (n = 4) and that the results should be regarded as preliminary and exploratory rather than generalizable across all lung cancer subtypes. And, we highlighted the inherent field-of-view limitations of the BL-14B beamline, which restrict whole-organ imaging and currently limit the ability to visualize tumor architecture in a broader anatomical context. Moreover, we describe the practical constraint of prolonged acquisition times, which remain a challenge for routine application. Finally, we emphasize that analyzer-based synchrotron microtomography is limited to ex-vivo imaging; its physical and radiation-dose constraints preclude any direct in-vivo translation at present.
Response 3
In the section of Discussion, we inserted the limitation from a small study population of this study according to the reviewer’s comment. New descriptions were marked with yellowish color in the text.
Certain considerations should be taken into account when interpreting the outcomes of this study. Foremost, the sample size was small (n = 4), and the findings should therefore be interpreted as preliminary and exploratory rather than representative of all lung cancer subtypes. The restricted field of view of the BL-14B imaging geometry also limits whole-organ visualization and prevents comprehensive mapping of tumor architecture in broader anatomic context. In addition, the relatively long acquisition times required for analyzer-based refraction-contrast imaging currently pose practical constraints on scalability. Moreover, this modality is inherently limited to ex-vivo application; radiation dose, beamline infrastructure, and optical configuration preclude its direct implementation in in-vivo settings.
Comment 4.
Please provide a more structured comparison with other synchrotron-based phase-contrast modalities. The discussion focuses heavily on facility descriptions (ESRF, SYRMEP, IMBL). Comparisons should shift toward technical distinctions and practical advantages/limitations of XDFI versus alternative phase-contrast approaches.
Answer 4
We thank the reviewer for this constructive suggestion. We agree that the Discussion should emphasize technical distinctions among phase-contrast imaging modalities rather than descriptive comparisons of synchrotron facilities. Accordingly, we have revised the Discussion to focus on the fundamental differences between crystal analyzer–based X-ray dark-field imaging (XDFI) and other synchrotron-based phase-contrast approaches, including propagation-based and grating-based techniques. The revised text highlights differences in contrast mechanisms, spatial resolution, sensitivity to microstructural interfaces, field-of-view scalability, and translational feasibility. While recent studies at ESRF, SYRMEP, and IMBL demonstrate multiscale, multilayer high-resolution imaging enabled by advanced accelerator infrastructures, we clarify that the present study emphasizes how dedicated optical design and imaging strategies can partially compensate for facility-level limitations. We further discuss how this approach provides methodological insights relevant to synchrotron centers without next-generation sources, while outlining future directions required to achieve multiscale and in-vivo-capable imaging comparable to leading facilities.
Response 4
We added sentences regarding the comparison between Synchrotron Facilities and implication of our studies in the section of Discussion (page__ line __) as the reviewer’s comment. Corrections were highlighted with yellowish color in the text.
All those recent advances in synchrotron radiation based biomedical researches have increasingly emphasized multiscale and multilayer three-dimensional visualization of biological tissues as well as clinical potentials supported by exponentially brilliant light source, in addition to extended long experimental facilities. Notably, ESRF operates a fourth-generation synchrotron source, while SYRMEP and IMBL are based on advanced third-generation accelerators, with ESRF and IMBL equipped with dedicated medical long beamlines and SYRMEP actively pursuing long-beamline upgrades.
However, beyond facility infrastructure, meaningful distinctions among phase-contrast approaches arise from their underlying contrast mechanisms and optical configurations. Propagation-based phase-contrast imaging excels in large field-of-view imaging and is well suited for multiscale and translational studies, but its contrast is strongly dependent on propagation distance and often requires phase-retrieval assumptions. Grating-based interferometric techniques provide quantitative phase information but face limitations in spatial resolution, dose efficiency, and scalability to thick or highly heterogeneous specimens.
In contrast, crystal analyzer–based X-ray dark-field imaging (XDFI), as employed in this study, offers exceptional sensitivity to subtle refraction-angle variations at air–tissue and tissue–tissue interfaces, enabling high-contrast visualization of alveolar microarchitecture and fibrotic remodeling without exogenous contrast agents. Importantly, the present work demonstrates that carefully optimized optical design and reconstruction strategies can partially compensate for the inherent constraints of a second-generation synchrotron facility such as the Photon Factory. This highlights a methodological pathway by which centers lacking next-generation accelerator infrastructure may still achieve biologically meaningful, high-resolution three-dimensional virtual histology.
Nevertheless, to remain competitive with state-of-the-art facilities and to advance toward clinical translation, future efforts must extend XDFI toward true multiscale, multilayer imaging with expanded fields of view and accelerated acquisition, ultimately enabling organ-scale coverage and, potentially, in-vivo applicability. These directions align with ongoing developments at ESRF, SYRMEP, and IMBL and define a clear roadmap for the evolution of analyzer-based phase-contrast tomography.
Comment 5
Please standardize arrow colors, labels, and scale bars across all figures to improve readability.
Answer 5
We deeply appreciate the reviewer’s valuable advice. Also, we sincerely apologize for the confusing use of arrow colors. We have revised all main and supplementary figures to standardize arrow colors, labels, and scale bars for improved clarity and consistency.
Across all figures, arrow colors are now uniformly defined as follows: blue indicates viable tumor components, yellow indicates necrotic, fibrotic, or keratinized regions, and red meant hemorrhagic congestion. For lepidic pattern which is characteristic in specimen #1, green color was used. Purple, for intratumor cleft in specimen #4, and bright yellow marked well defined peripheral stromal rim in specimen #4. This color scheme is applied consistently to both synchrotron radiation (SR) images and corresponding histological sections to facilitate direct visual comparison.
In addition, scale bars and panel labels have been harmonized across all figures, and corresponding descriptions have been added to the figure captions. These revisions improve figure readability and enhance correspondence between SR imaging and histopathology.
Comment 6
Please improve the resolution of Supplementary Figures S2–S5, where some labels are difficult to interpret.
Answer 6
Thank you very much for the precious comment. And we are very sorry for the poor resolution of supplementary figures. In response, we have regenerated all Supplementary Figures S2–S5 using the original high-resolution image data.
The resolution of these figures has been increased, and all labels, arrows, and annotations have been enlarged and repositioned to ensure clear legibility upon zooming in the online version. In addition, inset panels were re-cropped directly from the original datasets to avoid loss of detail during scaling. The revised Supplementary materials were submitted and we also provided high resolution images to the editorial office in cases of further improvement.
Comment 7
Please ensure consistent terminology (e.g., “SR-uCT,” “XDFI tomography,” “refraction-contrast imaging”).
Answer 7
I and my coauthors are truly grateful for the reviewer’s comment. We totally agree that we should correct the inconsistency of terminology such as SR-uCT, XDFI tomography, and refraction-contrast imaging for improving clarity and readability. We have tried to standardize those terminologies throughout the whole manuscript as “XDFI tomography (X-ray Dark-Field Imaging Tomography)” across all sections, including the main text, figure captions, and supplementary materials. Terms such as “SR-uCT,” “XDFI tomography,” and “refraction-contrast imaging” are now used only when necessary for explanatory purposes at first mention, or have been replaced to maintain terminological consistency. This revision ensures a clear and uniform description of the imaging modality throughout the manuscript.
Comment 8
Please add inset boxes on histology figures indicating precisely which region corresponds to the SR slice.
Answer 8
We would like to express our sincere gratitude for your valuable suggestion. Indeed, we really agree that our manuscript need clear spatial correspondence between synchrotron radiation tomographic imaging and pathologic examination.
In this study, all SR images and corresponding histologic figures shown in the main text represent magnified regions of interest selected from the originally acquired whole-specimen datasets. To clarify this correspondence, we have added blue inset boxes in the Appendix figures to indicate precisely which regions were enlarged and presented in the main figures.
The figures in Appendix were corrected to include the full-field SR images and corresponding histopathological slides, with blue boxes marking the locations corresponding to each magnified image shown in the main text. In addition, the figure legends have been revised to explicitly state that the main figures represent selected regions from the full datasets, which are provided in the Appendix for reference.
Comment 9
Please review formatting and units (pixel size, thickness, beam parameters) for consistency.
Answer 9
We thank the reviewer for highlighting the importance of consistent formatting and unit presentation. The manuscript has been carefully reviewed to ensure uniformity in the formatting and reporting of all imaging parameters.
Units for pixel size, specimen thickness, X-ray energy, beam geometry, and related parameters have been standardized throughout the main text, tables, figure captions, and supplementary materials using consistent SI unit notation. In addition, numerical values describing identical parameters were cross-checked across sections to ensure consistency and clarity. Those correction will be very helpful for improving the overall readability and technical precision of our manuscript.
Response to Comments on the Quality of English Language
Comments on the Quality of English Language
Please revise the manuscript carefully for English clarity, grammar, and readability. Below are specific areas that would benefit from improvement:
- Many sentences combine multiple ideas without appropriate punctuation, making the meaning difficult to parse. For example, in the Introduction and Discussion, several sentences exceed 40–50 words and include multiple subordinate clauses.
Please break these into shorter, more direct statements. - Certain concepts, such as contrast mechanisms, alveolar microstructures, and the advantages of synchrotron radiation, are repeated multiple times with slightly different wording. Please consolidate repetitive descriptions to enhance readability and avoid redundancy.
- Some sections switch between past and present tense when describing the same experiment or result. Please standardize to past tense for methods and results, and reserve present tense for interpretation or established knowledge.
- Terms such as “enhanced signals,” “grayish materials,” “porosity interruptions,” and “bright lines” need clearer definition or more standardized scientific wording. Please clarify these expressions to ensure precise communication. Examples include: “multiple sheet-like, mildly thickened alveolar walls”. “compact multi-layered band structures with rare porosity”. “low-attenuated dots-like structures”. These phrases are understandable but not idiomatic English. Please revise for scientific clarity and more natural expression.
- Descriptions of imaging findings often lack parallel structure. When comparing SR images to histology, the syntax becomes uneven and difficult to follow. Please use consistent sentence structure when describing paired findings (e.g., SR → histology → interpretation).
- Some transitions between sentences and paragraphs are abrupt. For example, in the Results section, transitions between tumor subtypes are sudden. Please add short linking sentences to guide the reader from one idea to the next.
- Some grammatical corrections are needed: Examples include: “identified as multiple sheet-like…” (should be “were identified as…”). “interruption within the fibrotic plate are spatially concordant…” (subject–verb agreement issue). “resemble to the fibrotic tumor bed” (should be “resemble the…”). Please revise these grammatical inconsistencies.
- Please correct the followings: missing spaces (“3Dreconstruction”), inconsistent hyphenation (“tumor-normal parenchymal margin”). Please correct typographical inconsistencies throughout.
- Improve figure captions for readability. Many captions contain long blocks of text with multiple findings in a single sentence. Please break them into shorter, structured sentences (e.g., describing normal, malignant, transitional zones separately).
Response
We sincerely appreciate for the reviewer’s thorough and detailed comments regarding English clarity, grammar, and overall readability of the manuscript. We fully agree that clear and precise language is essential for accurate scientific communication, and we greatly appreciate the reviewer’s careful attention to these aspects.
In response, we have comprehensively revised the manuscript in accordance with all points raised. Specifically, overly long and complex sentences have been divided into shorter, more direct statements; repetitive descriptions of contrast mechanisms, alveolar microstructures, and synchrotron-related advantages have been consolidated; and verb tense usage has been standardized, with past tense applied consistently to Methods and Results and present tense reserved for established knowledge and interpretation.
In addition, non-idiomatic or ambiguous expressions describing imaging findings have been replaced with more standardized scientific terminology, and parallel sentence structures have been adopted when describing corresponding synchrotron images and histopathological findings. Transitions between sections and between different tumor subtypes in the Results have been smoothed by adding brief linking sentences to improve narrative flow. Grammatical inconsistencies, typographical errors, spacing issues, and inconsistent hyphenation have also been carefully corrected throughout the manuscript. Figure captions were rewritten to improve readability by separating multiple findings into shorter, structured sentences.
To further ensure linguistic accuracy and consistency, the revised manuscript will undergo an additional round of review by a professional scientific English editor. We believe that the reviewer’s detailed suggestions have significantly improved the clarity, precision, and overall quality of the manuscript, and we are very grateful for the time and effort invested in providing these constructive comments.
The authors sincerely appreciate the editor’s and reviewers’ valuable comments. We carefully revised our manuscript according to the reviewers’ precious advice. We believe we did our best to improve the quality of our manuscript, and wish our revision have better achievement. If there were anything to be corrected or appended, please let us know and we promise we will make every effort to revise again.

Reviewer 2 Report
Comments and Suggestions for Authors
General comments:
The paper presents a unique imaging strategy with high scientific potential and well-documented technical achievements, but the current form is too long, unnecessarily descriptive, and lacks emphasis on the essential aims.
Clarifying the objectives, minimizing repetition, and polishing the methodological and clinical narratives would significantly increase clarity and effect.
Overall, the work is valuable, although it needs structural modification to improve readability and emphasize its significance.
Specific comments:
ABSTRACT
The abstract is extremely thick and technical; reduce and focus more clearly on the main goal, approach, and major findings.
Specify the number of samples and their subtypes to improve scientific transparency
NTRODUCTION
The introduction is detailed but overly long, with repeated discussions of CT limits and phase-contrast imaging; combine overlapping paragraphs to shorten it.
Clarify the actual information gap, such as standard CT's inadequate capacity to reveal microstructural tumour boundaries and subtype-specific architecture.
Increase the case for synchrotron-based dark-field imaging by illustrating why current PCI approaches are insufficient for lung cancer subtype discrimination.
To help the reader, state the exact purpose and hypothesis toward the end.
MATERIALS AND METHODS
Provide a more detailed explanation of why the four specific tumor specimens were chosen, as well as any limitations or representativeness.
The imaging setup description is extremely complex; consider summarizing major components and reserving comprehensive optical physics for extra resources.
To ensure clarity, specify voxel size, spatial resolution, total acquisition time, and reconstruction parameters in a single table.
Include further information about sample orientation and consistency between SR images and histology.
Improve organization by clearly separating subsections (sample preparation, imaging parameters, reconstruction, histopathology validation).
RESULTS
The results are visually appealing but overly descriptive. To improve clarity, try summarizing major findings by specimen in an organized format (e.g., bullet points or tables).
Reduce the number of comparable statements about alveolar structure preservation and tumor-normal interface appearance.
To establish uniformity, clarify imaging feature nomenclature (for example, "high attenuation," "refractive gradient," and "low-porosity domain").
Strengthen the relationship between imaging findings and their pathological correlates (e.g., necrosis, fibrosis, keratinization), perhaps in a summary figure/table.
Improve figure legends by providing brief explanations rather than lengthy paragraphs.
DISCUSSION
The discussion is overly lengthy and covers thorough evaluations of various synchrotron facilities; reduce and focus mostly on:
Key findings of this study include:
How XDFI tomography Complements Pathology
Advantages and restrictions unique to this dataset
Improve the clarity of constraints (for example, insufficient resolution for micropapillary structures or STAS, limited FOV).
Improve the clinical relevance section by outlining realistic future avenues to translational application.
Consider restructuring the conversation into subheadings to improve readability.
CONCLUSION
The conclusion is written in the style of a viewpoint piece, with only the most important outcomes being summarized and restated.
Avoid making sweeping generalizations about precision oncology unless they are specifically supported by the data.
Clearly indicate how this study advances the field and what further steps are required.
Several phrases in the manuscript are excessively long and contain multiple clauses, which reduces clarity and makes the narrative difficult to follow.
Redundant terminology appears in both the introduction and the discussion, resulting in repetition and loss of concentration.
Terminology is rarely regularly applied, particularly when describing imaging characteristics and microstructural patterns.
Some paragraphs should be restructured to guarantee logical flow and prevent combining techniques, findings, and interpretation.
Examples:
-
Overly long sentence: A sentence in the Introduction spans more than 5 lines and combines background, aim, and justification without separation, making it difficult to interpret.
-
Inconsistent terminology: The Results section alternates between “high-attenuation domain,” “dense region,” and “low-porosity zone” to describe similar imaging observations, which may confuse readers.
Author Response
We would like to sincerely appreciate the reviewer for the careful and constructive evaluation of our manuscript. We appreciate the reviewer’s recognition of the scientific potential and technical achievements of our work, as well as the insightful suggestions aimed at improving clarity, structure, and readability. In response, we have substantially revised the manuscript with particular attention to organization, conciseness, and consistency, while preserving the scientific content and references. Detailed responses to each comment are provided below.
Comments and Suggestions for Authors
Point-by-point response to Comments and Suggestions for Authors
General comments:
The paper presents a unique imaging strategy with high scientific potential and well-documented technical achievements, but the current form is too long, unnecessarily descriptive, and lacks emphasis on the essential aims.
Clarifying the objectives, minimizing repetition, and polishing the methodological and clinical narratives would significantly increase clarity and effect.
Overall, the work is valuable, although it needs structural modification to improve readability and emphasize its significance.
Response:
I and my coauthors are sincerely grateful for the reviewer’s valuable comment, and we totally agree with the assessment. The manuscript has been extensively restructured to improve readability and focus. Redundant descriptions have been removed or consolidated, and each section has been revised to emphasize the central objectives and key findings. In particular, the Introduction, Results, and Discussion sections were streamlined, and clearer structural separation was introduced across all major sections.
Specific comments:
Comment 1
ABSTRACT
The abstract is extremely thick and technical; reduce and focus more clearly on the main goal, approach, and major findings.
Specify the number of samples and their subtypes to improve scientific transparency
Response 1
Thank you for the precise advice. Also, we are fully aware of the need for abstract reconstruction. The Abstract has been fully revised to reduce technical detail and to clearly state the main objective, approach, and principal findings. The number of specimens (n = 4) and their histological subtypes are now explicitly stated to improve transparency and readability.
Comment 2
INTRODUCTION
The introduction is detailed but overly long, with repeated discussions of CT limits and phase-contrast imaging; combine overlapping paragraphs to shorten it.
Clarify the actual information gap, such as standard CT's inadequate capacity to reveal microstructural tumour boundaries and subtype-specific architecture.
Increase the case for synchrotron-based dark-field imaging by illustrating why current PCI approaches are insufficient for lung cancer subtype discrimination.
To help the reader, state the exact purpose and hypothesis toward the end.
Response 2
We are thoroughly grateful for the reviewer’s thoughtful advice. According to the reviewer’s comment, we have shortened and reorganized the Introduction by merging overlapping discussions of conventional CT and phase-contrast imaging. The information gap—namely, the limited ability of standard CT and existing PCI approaches to resolve subtype-specific lung microarchitecture—has been clarified. The final paragraph now explicitly states the purpose of the study and its exploratory hypothesis.
Comment 3
MATERIALS AND METHODS
Provide a more detailed explanation of why the four specific tumor specimens were chosen, as well as any limitations or representativeness.
The imaging setup description is extremely complex; consider summarizing major components and reserving comprehensive optical physics for extra resources.
To ensure clarity, specify voxel size, spatial resolution, total acquisition time, and reconstruction parameters in a single table.
Include further information about sample orientation and consistency between SR images and histology.
Improve organization by clearly separating subsections (sample preparation, imaging parameters, reconstruction, histopathology validation).
Response 3
Thank you very much for the reviewer’s valuable comment. The Materials and Methods section was comprehensively reorganized into clearly defined subsections (specimen selection and preparation, imaging system, acquisition and reconstruction, and histopathological correlation) as the reviewer had pointed out.
- The rationale for selecting four specific tumor specimens is now explicitly described, and the exploratory nature and limited representativeness are clearly acknowledged.
- The imaging setup description was simplified to highlight major components, with detailed optical physics moved to the Supplementary Materials.
- Key imaging and reconstruction parameters (voxel size, spatial resolution, field of view, acquisition time) are now consolidated into a single table.
- Additional information regarding specimen orientation and spatial correspondence between synchrotron images and histology has been added.
Comment 4
RESULTS
The results are visually appealing but overly descriptive. To improve clarity, try summarizing major findings by specimen in an organized format (e.g., bullet points or tables). Reduce the number of comparable statements about alveolar structure preservation and tumor-normal interface appearance. To establish uniformity, clarify imaging feature nomenclature (for example, "high attenuation," "refractive gradient," and "low-porosity domain"). Strengthen the relationship between imaging findings and their pathological correlates (e.g., necrosis, fibrosis, keratinization), perhaps in a summary figure/table. Improve figure legends by providing brief explanations rather than lengthy paragraphs.
Response 4
Thank you very much for the reviewer’s important comment. As the reviewer had pointed out, we should reduce and summarize duplicated and redundant sentences in this section. The Results section has been substantially revised to reduce descriptive redundancy and to emphasize specimen-specific findings. Each specimen is now presented in a concise, structured format focusing on key imaging–pathology correlations. Terminology describing imaging features (e.g., “high attenuation,” “low porosity,” “refractive gradient”) has been standardized throughout. Figure legends were shortened and revised to provide clear but succinct explanations without duplicating main-text descriptions.
Comment 5
DISCUSSION
The discussion is overly lengthy and covers thorough evaluations of various synchrotron facilities; reduce and focus mostly on:
Key findings of this study include:
How XDFI tomography Complements Pathology
Advantages and restrictions unique to this dataset
Improve the clarity of constraints (for example, insufficient resolution for micropapillary structures or STAS, limited FOV).
Improve the clinical relevance section by outlining realistic future avenues to translational application.
Consider restructuring the conversation into subheadings to improve readability.
Response 5
We sincerely thank the reviewer for this constructive and insightful comment. We agree that the original Discussion placed excessive emphasis on descriptive comparisons of synchrotron facilities and that a more focused and structured presentation would improve clarity and impact.
In response, we have substantially revised and reorganized the Discussion section. The revised Discussion is now structured around clear subheadings to improve readability and to directly address the reviewer’s recommendations. Specifically, we have refocused the Discussion on: (1) the key findings of the present study, (2) how synchrotron-based XDFI tomography complements conventional histopathology, (3) advantages and restrictions unique to the current dataset, and (4) realistic future directions toward translational application.
Detailed descriptions of individual synchrotron facilities have been reduced and reframed to provide methodological context rather than exhaustive comparison, while still retaining essential contrasts as requested by Reviewer 1. The limitations of the present approach—including insufficient spatial resolution for micropapillary components and spread through air spaces (STAS), restricted field of view, long acquisition times, and current restriction to ex vivo imaging—are now stated more explicitly and critically.
Furthermore, the clinical relevance section has been revised to avoid overstatement. Rather than proposing immediate clinical implementation, we now outline realistic and stepwise future avenues, emphasizing improvements in spatial resolution, field-of-view expansion, quantitative validation, and pathology-adjacent research applications as necessary prerequisites for broader translational relevance.
We believe that these revisions have significantly improved the focus, balance, and readability of the Discussion, and more clearly position the present work as an exploratory, methodologically oriented study with defined strengths, limitations, and future directions.
Comment 6
CONCLUSION
The conclusion is written in the style of a viewpoint piece, with only the most important outcomes being summarized and restated.
Avoid making sweeping generalizations about precision oncology unless they are specifically supported by the data.
Clearly indicate how this study advances the field and what further steps are required.
Response 6
I and my coauthors are truly grateful for the reviewer’s thoughtful and helpful comment. We agree that the original Conclusion may have conveyed a viewpoint-oriented tone and that a more concise and balanced summary was warranted.
In response, we have carefully revised the Conclusion to focus exclusively on the principal findings and contributions of the present study. Broad or speculative statements regarding precision oncology have been removed or appropriately moderated to ensure that all claims are directly supported by the data. The revised Conclusion now more clearly articulates how this work contributes to the development of three-dimensional virtual histology in lung cancer, while also explicitly outlining the methodological and technical steps required for future advancement.
We hope that these revisions have improved the clarity, balance, and appropriateness of the Conclusion and better align it with the scope and exploratory nature of the present study.
Response to Comments on the Quality of English Language
Comments on the Quality of English Language
Several phrases in the manuscript are excessively long and contain multiple clauses, which reduces clarity and makes the narrative difficult to follow.
Redundant terminology appears in both the introduction and the discussion, resulting in repetition and loss of concentration.
Terminology is rarely regularly applied, particularly when describing imaging characteristics and microstructural patterns.
Some paragraphs should be restructured to guarantee logical flow and prevent combining techniques, findings, and interpretation.
Examples:
- Overly long sentence: A sentence in the Introduction spans more than 5 lines and combines background, aim, and justification without separation, making it difficult to interpret.
- Inconsistent terminology: The Results section alternates between “high-attenuation domain,” “dense region,” and “low-porosity zone” to describe similar imaging observations, which may confuse readers.
Response:
We indeed appreciate for the reviewer’s detailed and constructive comments regarding language clarity, structural organization, and terminological consistency. We fully agree that overly long sentences, redundancy, and inconsistent terminology can hinder readability and obscure the intended scientific message.
In response, we have carefully revised the manuscript to address all of these concerns. Overly long sentences—particularly in the Introduction—have been divided into shorter, more focused statements that separately present background, study rationale, and objectives. Redundant descriptions appearing in both the Introduction and Discussion have been consolidated to reduce repetition and improve narrative coherence.
Furthermore, terminology describing imaging characteristics and microstructural patterns has been standardized throughout the manuscript. Expressions such as “high-attenuation domain,” “dense region,” and “low-porosity zone” have been unified under consistent, clearly defined terminology to avoid confusion. Paragraphs in which methodological details, results, and interpretation were previously intermingled have been restructured to ensure a clearer logical progression and to improve readability.
We believe that these revisions have significantly enhanced the clarity, consistency, and overall flow of the manuscript, and we are grateful to the reviewer for these valuable suggestions.
We again thank the reviewer for the thoughtful and constructive feedback, which has significantly improved the clarity, structure, and overall quality of the manuscript. We believe that the revised version addresses all concerns and more clearly communicates the significance and limitations of this exploratory study.

Reviewer 3 Report
Comments and Suggestions for Authors
This manuscript investigates the use of synchrotron radiation–based X-ray dark-field imaging and refraction-contrast tomography to visualize microstructural features of various lung cancer tissues. the study provides interesting high-resolution imaging of alveolar structures and compares them with histopathology. The topic is scientifically important and clinically relevant, specifically in the context of early lung cancer diagnosis. However, the manuscript requires substantial revisions in methodology, quantitative analysis, clarity, and presentation before it can be considered for publication.
- The study relies almost entirely on qualitative descriptions. No quantitative measurements are provided to objectively support the claimed imaging advantages. Please include quantitative metrics that you think it is suitable such as contrast-to-noise ratio, signal-to-noise ratio, alveolar wall thickness, lumen size, porosity attenuation/phase-shift values segmentation performance inter-reader agreement between pathologists. Quantitative evidence is essential to demonstrate diagnostic value.
- Only four specimens are included, yet conclusions are generalized to broad lung cancer subtypes. Please justify the sample size and clearly state that results are preliminary and exploratory. The manuscript should avoid overgeneralization.
- Several critical steps of the imaging and reconstruction workflow are not explained sufficiently to enable reproducibility. For example, rationale for 19.8 keV energy selection, reconstruction algorithm and parameters.
- In Figure A1 to A4, what the difference colour of the arrow means/indicate? Please add explanation to the figure caption to enhance the clarity and readability
- Synchrotron phase-contrast and refraction-based micro-CT have been previously reported. The manuscript does not clearly identify what is new in this work. what XDFI configuration or technique is new? what advantage it provides over prior synchrotron μCT studies? why this approach is unique for lung cancer?
- The discussion is long but does not address key translational issues. Please discuss the radiation dose, synchrotron availability, comparison with commercial micro-CT, cost and scalability, potential workflow integration into pathology, and the limitations for clinical implementation.
- The current limitations section is descriptive but not critical. You can explicitly acknowledge the small sample size, lack of statistical analysis aand absence of quantitative validation
Author Response

(The authors gave the same response as above.)

Round 2
Reviewer 1 Report
Comments and Suggestions for Authors
The authors have submitted the revisions and adequately addressed my questions. I thank them for their thorough responses and have no further comments at this time.
Comments on the Quality of English LanguageThe authors have submitted the revisions and adequately addressed my questions. I thank them for their thorough responses and have no further comments at this time.
Author Response
|
Response to Reviewer’s Comments
|
We would like to express our sincere gratitude for the reviewer’s careful re-evaluation of our manuscript and for the positive feedback. We believe that the manuscript has been meaningfully improved in quality as a result of the reviewer’s insightful comments and dedication. Once again, we sincerely appreciate the reviewer’s valuable time and effort.
Comments and Suggestions for Authors
Comment 1
- The authors have submitted the revisions and adequately addressed my questions. I thank them for their thorough responses and have no further comments at this time.
Answer 1
We would like to extend our deepest gratitude to the reviewer for their insightful evaluation and for confirming that no further revisions are necessary. We truly appreciate the time and effort dedicated to reviewing our manuscript. The constructive guidance provided throughout the review process has been invaluable in refining the quality and clarity of our work.
Comments on the Quality of English Language
Comment 1
- The authors have submitted the revisions and adequately addressed my questions. I thank them for their thorough responses and have no further comments at this time.
Answer 1
We appreciate the reviewer’s acknowledgment that the revisions adequately addressed the previous concerns. We thank the reviewer for their valuable time and have no further comments at this stage.
The authors sincerely appreciate the reviewers’ valuable comments. We carefully revised our manuscript according to the reviewers’ precious advice. We believe we did our best to improve the quality of our manuscript, and wish our revision have better achievement. If there were anything to be corrected or appended, please let us know and we promise we will make every effort to revise again.

Reviewer 3 Report
Comments and Suggestions for Authors
The authors have not adequately responded to my original comments. Instead, they appear to address different points, making it difficult for me to assess the revisions and determine how (or whether) my concerns have been resolved
Author Response
Response to Reviewer's Comments
We would like to sincerely appreciate the reviewer for the careful and constructive evaluation of our manuscript. We appreciate the reviewer’s recognition of the scientific potential and technical achievements of our work, as well as the insightful suggestions aimed at improving clarity, structure, and readability. In response, we have substantially revised the manuscript with particular attention to organization, conciseness, and consistency, while preserving the scientific content and references. Detailed responses to each comment are provided below.
Round 2. Comments and Suggestions for Authors
Comment 1
The authors have not adequately responded to my original comments. Instead, they appear to address different points, making it difficult for me to assess the revisions and determine how (or whether) my concerns have been resolved
Answer 1
We sincerely apologize to the reviewer for the confusion caused by our previous response. Upon careful re-examination, we recognize that our reply in the previous round did not adequately address the reviewer’s original comments and instead responded to different points. This was an unintentional oversight on our part.
We truly apologize for this terrible mistake, and greatly appreciate the reviewer’s patience. We would like to take this opportunity to provide a clear and direct response to each of the original concerns raised. Below, we have addressed the reviewer’s comments point by point, explicitly indicating how each concern has been considered and revised in the manuscript.
We hope that the revised responses and corresponding changes will allow the reviewer to more easily assess how the concerns have now been fully addressed. We sincerely thank the reviewer again for the careful evaluation and valuable feedback.
Round 1. Comments and Suggestions for Authors
Point-by-point response to Comments and Suggestions for Authors
Comments and Suggestions for Authors
This manuscript investigates the use of synchrotron radiation–based X-ray dark-field imaging and refraction-contrast tomography to visualize microstructural features of various lung cancer tissues. the study provides interesting high-resolution imaging of alveolar structures and compares them with histopathology. The topic is scientifically important and clinically relevant, specifically in the context of early lung cancer diagnosis. However, the manuscript requires substantial revisions in methodology, quantitative analysis, clarity, and presentation before it can be considered for publication.
Comment 1
- The study relies almost entirely on qualitative descriptions. No quantitative measurements are provided to objectively support the claimed imaging advantages. Please include quantitative metrics that you think it is suitable such as contrast-to-noise ratio, signal-to-noise ratio, alveolar wall thickness, lumen size, porosity attenuation/phase-shift values segmentation performance inter-reader agreement between pathologists. Quantitative evidence is essential to demonstrate diagnostic value.
Answer 1
We are truly grateful for the reviewer’s precise comment. We totally agree that the quantitative analysis including metrics such as signal intensity profiles, porosity maps, or refractive-index contrast measurements should be mentioned, and would like to deeply apologize that we have not described any of quantitative metrics.
In this study, we focused on the qualitative comparison between refraction-contrast images using synchrotron radiation and pathologic examinations. Quantitative metrics—such as signal intensity profiles, porosity distributions, or refractive-index contrast measurements—were not incorporated because voxel-level registration between SR datasets and histologic sections remains technically challenging in heterogeneous lung specimens. The intrinsic tissue deformation that occurs during formalin fixation, paraffin embedding, and sectioning prevents accurate one-to-one correspondence with the intact three-dimensional SR volume, thereby limiting the feasibility of deriving reliable quantitative microstructural measurements within the scope of this feasibility study.
We are thoroughly aware that quantitative validation will be essential those kinds of studies and would like to compensate this weakness in the subsequent research work. We have therefore revised the manuscript to explicitly discuss this limitation and outline a future methodological framework that includes voxel-to-slide co-registration, automated segmentation pipelines, and quantitative extraction of structural parameters such as alveolar wall thickness, acinar porosity, phase-edge amplitude, and necrotic burden. These developments will allow more robust statistical assessment and enhance the diagnostic utility of SR-based virtual histology in the following our studies.
Response 1
We would like to add paragraphs describing that the absence of quantitative analysis was one of main limitation with our current study in the Discussion section, according to the reviewer had pointed out. Corrections were highlighted with yellowish color in the text.
Another essential limitation of the present study is the absence of quantitative analysis. Although qualitative correspondence between synchrotron refraction-contrast imaging and histopathology provides meaningful preliminary insight, reliable voxel-to-slide co-registration could not be achieved. Lung tissue is particularly susceptible to fixation-related deformation and anisotropic shrinkage, which complicate precise spatial matching between synchrotron datasets and histological sections. Consequently, quantitative microarchitectural validation could not be incorporated into the current investigation.
We consider this limitation as a critical area for further development and are actively pursuing follow-up studies aimed at establishing reproducible quantitative indices that integrate refraction-contrast signatures with pathological ground truth. These efforts include the development of metrics related to alveolar wall morphology, phase-edge sharpness, acinar porosity, and treatment-related stromal alterations, which are expected to form the methodological foundation for future extensions of this work.
Comment 2
- Only four specimens are included, yet conclusions are generalized to broad lung cancer subtypes. Please justify the sample size and clearly state that results are preliminary and exploratory. The manuscript should avoid overgeneralization.
Answer 2
We sincerely thank the reviewer for this valuable advice. Indeed, we are fully aware that the limitation induced by a small study population should be mentioned more explicitly to help readers appropriately interpret the scope and implications of our findings. In response, we have revised the Discussion section to provide a clearer and more comprehensive description of the constraints associated with our current work.
In the revised discussion, we would like to explicitly acknowledge that the sample size is small (n = 4) and that the results should be regarded as preliminary and exploratory rather than generalizable across all lung cancer subtypes. And, we highlighted the inherent field-of-view limitations of the BL-14B beamline, which restrict whole-organ imaging and currently limit the ability to visualize tumor architecture in a broader anatomical context. Moreover, we describe the practical constraint of prolonged acquisition times, which remain a challenge for routine application. Finally, we emphasize that analyzer-based synchrotron microtomography is limited to ex-vivo imaging; its physical and radiation-dose constraints preclude any direct in-vivo translation at present.
Response 3
In the section of Discussion, we inserted the limitation from a small study population of this study according to the reviewer’s comment. New descriptions were marked with yellowish color in the text.
Certain considerations should be taken into account when interpreting the outcomes of this study. Foremost, the sample size was small (n = 4), and the findings should therefore be interpreted as preliminary and exploratory rather than representative of all lung cancer subtypes. The restricted field of view of the BL-14B imaging geometry also limits whole-organ visualization and prevents comprehensive mapping of tumor architecture in broader anatomic context. In addition, the relatively long acquisition times required for analyzer-based refraction-contrast imaging currently pose practical constraints on scalability. Moreover, this modality is inherently limited to ex-vivo application; radiation dose, beamline infrastructure, and optical configuration preclude its direct implementation in in-vivo settings.
Comment 3
- Several critical steps of the imaging and reconstruction workflow are not explained sufficiently to enable reproducibility. For example, rationale for 19.8 keV energy selection, reconstruction algorithm and parameters.
Answer 3
I and my coauthors sincerely appreciate for the reviewer’s thoughtful suggestion regarding the need for improving methodological clarification. We have added sentences for improving methodological clearance, regarding rationale for rationale for 19.8 keV energy selection and other experimental conditions.
In the present manuscript, however, our primary objective was to demonstrate the feasibility and imaging performance gains achieved through crystal analyzer–based optical optimization, particularly the improved spatial resolution and refraction-contrast sensitivity resulting from the modified XDFI configuration. For this reason, the reconstruction algorithm itself was intentionally not treated as a central variable in this study, and only standard reconstruction procedures routinely employed for analyzer-based refraction-contrast tomography were applied.
We are fully aware that the lack of a detailed discussion of reconstruction algorithms and parameter choices represents a limitation of the current manuscript. To avoid conflating optical hardware–driven improvements with algorithmic contributions, we chose to focus this work on crystal design, analyzer configuration, and imaging performance outcomes. A comprehensive methodological analysis of reconstruction strategies—including algorithmic variations, parameter sensitivity, and their quantitative impact on spatial resolution and contrast—is currently underway and will be addressed explicitly in a subsequent dedicated study.
To clarify this point for readers, we have revised the manuscript to better delineate the scope of the present work and to explicitly state that detailed reconstruction methodology will be reported separately. We believe that this separation allows for clearer attribution of the observed resolution improvements to optical advancements, while ensuring that algorithmic considerations are treated rigorously in future publications.
Response 3
We would like to insert sentences to supplement the insufficient explanation of the Materials and Methods section.
In 2.2. X-ray source and Experimental setup of part 2 Materials and Methods section. We would like to insert two additional paragraphs describing why we chose the incident X-ray energy of 19.8keV, and the optical configuration of Laue-Angle Analyzer (LAA). Changes were marked with yellowish color in the text.
An incident X-ray energy of 19.8 keV was applied to balance phase-contrast sensitivity and penetration depth for human lung soft tissues. At this energy, re-fraction angles generated by alveolar septa and tumor–stroma interfaces are sufficiently large to be detected by analyzer-based optics while minimizing absorption-related attenuation. Detailed optical physics of the analyzer-based configuration are provided in the Supplementary Materials (Figure S1).
The LAA consisted of 166-μm-thick Si (111) plates with a 5° asymmetric cut. This asymmetric Laue-case configuration increases angular dispersion and im-proves sensitivity to subtle electron-density gradients by modulating the crystal’s acceptance angle. As a result, small refraction angles produced at alveolar septa and tumor–stroma boundaries are selectively filtered and amplified, thereby enhancing phase sensitivity and soft-tissue contrast.
In 2.3. Imaging acquisition and Reconstruction of part 2 Materials and Methods section. We would like to insert two additional paragraphs describing (3) the way of these parameters affecting spatial resolution, phase sensitivity, and overall imaging performances. Newly inserted paragraphs were highlighted with yellowish color in the text.
The X-ray detector employed a camera with a physical pixel size of 2.75μm; however, the effective spatial resolution of the entire imaging system, including the synchrotron source properties, silicon single-crystal optics, scintillator in the detector, and geometric blurring, was approximately 10μm.
In the ligand of Supplementary figure A1, we would like to add the theoretical base of our experiment.
The incident synchrotron X-ray beam is first diffracted by the AMC under an asymmetric Bragg condition, generating a broadened and collimated exit beam whose divergence is governed by the asymmetric factor b = sin (θB−α)/sin (θB+α), where θB is the Bragg angle and α is the asymmetry angle of the crystal. As the beam propagates through the specimen, refraction and phase shifts cause angular deviations (φ) proportional to local electron-density gradients. Downstream, the beam is separated by the LAA into forward-diffraction (FD) and analyzer-diffraction (AD) paths, producing complementary refraction-contrast signals. The resulting FD and AD intensity profiles capture sub-microradian angular variations, enabling high-sensitivity phase-contrast imaging of soft-tissue microstructures. This geometric configuration forms the theoretical basis for the refraction-contrast microtomography used in this study. Forward-diffracted rays, captured by the X-ray camera, provided refraction-contrast images representing angular deviations due to specimen structure. The acrylic filter minimized artifacts arising from the shape of sample containers. Specimens were stabilized in cylindrical agarose-filled containers to avoid significant X-ray refraction occurring at the boundary between the specimen and the air, which leads to severe image artefacts.
Comment 4
- In Figure A1 to A4, what the difference colour of the arrow means/indicate? Please add explanation to the figure caption to enhance the clarity and readability
Answer 4
We thoroughly thank the reviewer for this valuable comment regarding figure clarity. The explanation for figures and refinement of resolution should be revised. And we totally agree that explicit explanation of arrow color coding is essential for accurate interpretation of the supplementary figures.
In response, we have revised the captions of Figures A1–A4 to clearly define the meaning of each arrow color. The color scheme has now been standardized and consistently applied across all main and supplementary figures. Specifically, arrow colors indicate the following:
- Green arrows; lepidic growth pattern (specific for adenocarcinoma)
- Blue arrows; viable tumor component
- Red arrows; hemorrhagic congestion
- Yellow arrows; solid or necrotic stroma
- Purple arrows; intra-tumoral cleft
- Bright yellow arrows; well-defined peripheral stromal rim
Comment 5
- Synchrotron phase-contrast and refraction-based micro-CT have been previously reported. The manuscript does not clearly identify what is new in this work. what XDFI configuration or technique is new? what advantage it provides over prior synchrotron μCT studies? why this approach is unique for lung cancer?
Response 5
I and my coauthors are sincerely grateful for the reviewer’s important and thoughtful comment. Also, we totally agree that synchrotron phase-contrast and refraction-based micro-CT techniques have been previously reported, and we appreciate the opportunity to clarify the novelty and specific contribution of the present work more explicitly.
We fully understand conventional absorption-based CT and many existing phase-contrast imaging (PCI) approaches remain limited in their ability to consistently resolve complex alveolar microarchitecture and tumor–parenchyma interfaces in three dimensions, particularly in lung tissue, where strong air–tissue heterogeneity dominates image formation. While propagation-based and grating-based PCI techniques have provided important advances in soft-tissue contrast, their performance in highly heterogeneous, air-containing organs is often constrained by propagation distance requirements, phase-retrieval assumptions, or reduced robustness across varying tissue compositions.
The present study does not aim to introduce a fundamentally new phase-contrast principle. Rather, its novelty lies in the careful application and optimization of a crystal analyzer–based X-ray Dark-Field Imaging (XDFI) configuration that combines dark-field detection with enhanced refraction sensitivity, specifically tailored to emphasize small-angle refraction and scattering at air–tissue and tissue–tissue interfaces. This optical sensitivity is particularly relevant for lung tissue, where subtle alterations in alveolar frameworks, fibrotic remodeling, keratinization, and treatment-related architectural changes play a critical pathological role.
Although XDFI has been previously applied to other soft tissues, such as the brain, breast, and eye, its systematic evaluation in human lung cancer specimens across different histological contexts—including post-chemoradiation changes and metastatic disease—has been limited. In this respect, the present study extends prior synchrotron μCT work by demonstrating that analyzer-based XDFI can reproducibly delineate subtype-associated microarchitectural features and tumor–normal transitional zones in lung cancer, a setting in which conventional PCI approaches often face intrinsic challenges.
We have revised the Introduction and Discussion to clarify these points according to the reviewer’s comment, and to more explicitly position the present work as an exploratory study demonstrating the suitability and potential advantages of crystal analyzer–based XDFI for three-dimensional virtual histology of lung cancer, while carefully acknowledging its current limitations.
Comment 6
- The discussion is long but does not address key translational issues. Please discuss the radiation dose, synchrotron availability, comparison with commercial micro-CT, cost and scalability, potential workflow integration into pathology, and the limitations for clinical implementation.
Response 6
We sincerely thank the reviewer for this thoughtful comment and fully appreciate the importance of clearly delineating the clinical limitations and translational context of the present approach.
We would like to respectfully emphasize that we are well aware of the substantial constraints that currently limit the direct clinical application of synchrotron-based XDFI CT, including radiation dose considerations, restricted accessibility to synchrotron facilities, long acquisition times, and scalability challenges. Accordingly, the present study was not designed to propose immediate clinical deployment. Rather, our primary objective was to investigate the potential of crystal analyzer–based XDFI for improving spatial resolution and expanding the effective imaging field of view in lung tissue, which represents a particularly challenging imaging environment due to its air–tissue heterogeneity.
With this perspective, we have revised the Discussion to more explicitly acknowledge both the methodological limitations of the technique and the resulting constraints on interpretation of the findings. We now clarify that the results should be interpreted within the context of an exploratory, ex vivo study focused on imaging performance rather than clinical feasibility. In addition, we have expanded the Discussion to address the limitations related to resolution, field of view, quantitative validation, and translational applicability, thereby providing a more balanced and realistic assessment of the current scope of the method.
We hope that these revisions more accurately convey our cautious and stepwise approach toward the development of XDFI CT, in which advances in resolution and imaging coverage are regarded as necessary prerequisites before any consideration of broader clinical translation.
Comment 7
- The current limitations section is descriptive but not critical. You can explicitly acknowledge the small sample size, lack of statistical analysis and absence of quantitative validation
Response
We sincerely thank the reviewer for this important and constructive comment. We fully agree that the limitations of the present study should be stated more explicitly and critically.
In the revised manuscript, we tried to describe more clearly relating with major limitations of our study; the small sample size, absence of statistical analysis, and lack of quantitative validation as the reviewer had pointed out. We emphasize that the present study is exploratory in nature and that the findings should therefore be interpreted with appropriate caution.
We would also like to respectfully note that we are fully aware of these limitations and regard them as key issues to be addressed in future investigations. In particular, we plan to expand the sample size, incorporate robust quantitative metrics, and refine methodological descriptions and analytical frameworks in subsequent studies. These improvements are intended to strengthen statistical rigor, enhance reproducibility, and enable more objective validation of imaging findings.
Through these efforts, we aim to build upon the present feasibility study and to advance toward more scientifically and biomedically meaningful investigations that better address the challenges identified in this review.
We again thank the reviewer for the thoughtful and constructive feedback, which has significantly improved the clarity, structure, and overall quality of the manuscript. We believe that the revised version addresses all concerns and more clearly communicates the significance and limitations of this exploratory study.
